# What Genetics Can Do for Oncological Imaging: A Systematic Review of the Genetic Validation Data Used in Radiomics Studies

**DOI:** 10.3390/ijms23126504

**Published:** 2022-06-10

**Authors:** Rebeca Mirón Mombiela, Anne Rix Arildskov, Frederik Jager Bruun, Lotte Harries Hasselbalch, Kristine Bærentz Holst, Sine Hvid Rasmussen, Consuelo Borrás

**Affiliations:** 1Radiology Derpartment, Herlev og Gentofte Hospital, Borgmester Ib Juuls Vej 17, Opgang 4, 4.Etage, E2, 2730 Herlev, Denmark; annerixarildskov@gmail.com (A.R.A.); jagerbruun@gmail.com (F.J.B.); lotte_hasselbalch@hotmail.com (L.H.H.); kristinebaerentzholst@hotmail.com (K.B.H.); sinehvid@gmail.com (S.H.R.); 2Freshage Research Group, Department of Physiology, Faculty of Medicine, University of Valencia, 46010 Valencia, Spain; 3CIBERFES, Institute of Health Research-INCLIVA, 46010 Valencia, Spain

**Keywords:** radiogenomics, radiomics, genomics, genetic validation, cancer, survival, prognosis, risk assessment, treatment response

## Abstract

(1) Background: Radiogenomics is motivated by the concept that biomedical images contain information that reflects underlying pathophysiology. This review focused on papers that used genetics to validate their radiomics models and outcomes and assess their contribution to this emerging field. (2) Methods: All original research with the words radiomics and genomics in English and performed in humans up to 31 January 2022, were identified on Medline and Embase. The quality of the studies was assessed with Radiomic Quality Score (RQS) and the Cochrane recommendation for diagnostic accuracy study Quality Assessment 2. (3) Results: 45 studies were included in our systematic review, and more than 50% were published in the last two years. The studies had a mean RQS of 12, and the studied tumors were very diverse. Up to 83% investigated the prognosis as the main outcome, with the rest focusing on response to treatment and risk assessment. Most applied either transcriptomics (54%) and/or genetics (35%) for genetic validation. (4) Conclusions: There is enough evidence to state that new science has emerged, focusing on establishing an association between radiological features and genomic/molecular expression to explain underlying disease mechanisms and enhance prognostic, risk assessment, and treatment response radiomics models in cancer patients.

## 1. Introduction

Imaging is an important technology in medical science, which is often used to aid decision-making. The usage of medical imaging is swiftly evolving from it being primarily a diagnostic tool to it playing a central role in the context of personalized medicine with the advent of radiomics in cancer patients [1]. Radiomics refers to the qualitative and quantitative extraction of data from clinical images, and the methodology used to convert imaging features in a way that supports decision-making [2]. It has been used in the field of oncology for outcome prediction, risk assessment, treatment response, tumor staging, and assessment of cancer genetics, also known as tumor phenotyping [3,4]. Since it first appeared in print in 2012, publications referring to “Radiomics” have increased exponentially [5], numbering over 5000 when the term is searched in PubMed in early 2022.

Radiomics is motivated by the concept that biomedical images contain information that reflects underlying pathophysiology, and these relationships can be revealed via quantitative image analysis that offers information on the disease microenvironment and the disease state [6]. Through direct quantification of the tumor imaging phenotype at the spatial scale within the resolution of the imaging technique used, radiomics aims to provide indirect insight into multiple aspects of the disease, from tumor grading to diagnosing the histologic and/or genetic subtype. These characteristics are reflective of alterations occurring at different spatial scales to the data provided by radiomics (Figure 1). Hence, the biological basis of the indirect relationships enabling radiomics predictions remains largely unexplained in most studies. Without an underlying biological rationale, the black box-like nature of “omics” methods significantly hinders its wider use and makes validation particularly challenging. Providing the biological context of the informative radiomics features will constitute an important step toward general acceptance of radiomics as a standalone diagnostic, predictive, or prognostic method in both the radiology and oncology communities [6]. Radiomics is not a panacea for clinical decision-making and there is increased pressure for robust radiomics [1], including the need to develop guidelines for standardized data collection, evaluation criteria, and reporting results. It is likely that biological validation will also become a standard practice in the field, thus helping translate basic science knowledge into clinical decision-making. This has led to a very specific science or types of studies where the link between specific imaging traits, and specific gene-expression patterns that inform the underlying cellular pathophysiology are studied. This science, known as radiogenomics, is actively expanding in the molecular oncology and clinical oncology fields, as cancer is a disease that involves genetic abnormalities caused by hereditary or environmental factors [2]. The main reason for this is that first, access to genomic information in conventional clinical procedures is based mainly on biopsy of focal tissue samples and microarray genetic analysis. Second, gene expression profiling of only a fraction of the tumor tissue cannot reflect the heterogeneity of the entire tumor [2], making imaging an interesting substitute for this purpose, something that is being called a virtual biopsy. Lastly, it has the potential that at multiple time points the optimal characterization of a tumor can be achieved [1].

There are multiple reviews dealing with radiomics and radiogenomics advances in many different cancer types. The most recent reviews on radiogenomics have focused on the state of the science [7], its capacity for making an accurate diagnosis [8], and its predictive value [2]. Zanfardino et al. (2019) published two reviews dealing with the subject. In one they focused on the different methodologies that can be used to integrate both genomics and radiomics into a multi-omics/multidisciplinary study [9], and in the second one, their group explore the impact of public genetic and imaging databases have on radiogenomics cancer research [9]. In addition, two reviews have focused on how deep learning and artificial intelligence are helping to advance this science [10,11].

This systematic review search, analysis, and selection focused on papers that have used genetics to validate their radiomics predictive model and outcomes in cancer patients; and assessed their contribution to this emerging field. Furthermore, understanding the biological basis of observed relationships, where possible, can strengthen conclusions and provide additional validation and opportunities for research.

## 2. Results

We identified 81 studies that met our inclusion criteria, and after quality assessment, 45 studies were included in our systematic review. The excluded studies were deemed to lack clarity regarding the radiomics process, recruitment, and selection of participants or there was missing information for the flow and timing of the clinical outcome and/or the genetical validation. The results for each stage of the search are demonstrated in the PRISMA flow diagram (Figure 2).

### 2.1. Study Characteristics

All studies were published between 2014 and 2021, with a little more than 50% published in the last two years (Figure 3). The included studies had a mean Radiomics Quality Score (RQS) of 12, which ranges from 5 to 20. Regarding the imaging modality 21 studies based their radiomics models on Computed Tomography (CT) images, 20 studies used magnetic resonance imaging (MRI), and 4 studies used positron emission tomography (PET) with CT.

The type of tumors found was very diverse. Fifteen articles studied brain tumors, eight studies focused on lung and head and neck tumors, six studies studied renal cell carcinoma, three studies investigated breast and ovarian cancer, respectively, and two studies focused on esophagus cancer. Endometrial cancer, bladder cancer, prostate cancer, gastric cancer, colorectal cancer, hepatocellular carcinoma, and melanoma had one study each. Lastly, one article dealt with several different types of solid tumors (Figure 4).

Eighty-three percent of the studies investigated the prognosis of the developed radiomic models, 17% investigated the risk assessment capabilities, and only 7% investigated treatment response. When the prognosis was tested, the most common outcome measure determined was overall survival (OS), which was tested in 33 of the included studies. Nine studies measured progression-free survival (PFS), three measured disease-free survival (DFS) and progression-free interval (PFI), respectively, and one study measured disease-specific survival (DSS), and recurrence-free survival (RFS), metastasis-free survival (MFS), and relapse, respectively. Eight studies evaluated the utilization of radiomics models for determining treatment response. Of those studies, four measured the response to treatment based on survival measurement, one study used hyperprogression or time-to-treat failures (TTF), one measured durable clinical benefit, and three studies determined response to treatment based on pathologic complete response (pCR). Risk assessment based on radiomic models was tested in three studies, one measured risk of recurrence, another measured risk of metastasis, and the other one measured early recurrence (Figure 5).

Most studies dealt with transcriptomics (54%) and/or genetics (35%), but 7% dealt with epigenomics and only 3.5% with proteomics. Twenty-eight studies used RNA sequencing or gene expression, 13 investigated specific mutation status and four determined the *O(6)-methylguanine-DNA methyltransferase promoter (MGMT)* methylation status, and whole-exome sequencing, respectively. One study evaluated the DNA copy number and one used Fluorescence In Situ Hybridization (FISH). Regarding the use of public genetics databases, 24 out of 45 (53%) used them. The most used databases were The Cancer Genome Atlas (TCGA) program and The Cancer Imaging Archive (TCIA), but several others were also used (Figure 6).

The size of the data sets and the number of patients included just for the genetic validation and the total sample of patients varies throughout the studies. The data set for the genetic validation ranged from 24 to 681, with a mean of 163 and a median of 91 patients. The number of patients included overall for all the analysis ranged from 24 to 1362, with a mean of 289 and a median of 153 patients.

The most pronounced limitation for the genetic validation of radiogenomics, as highlighted in Figure 7, is related to a lack of external validation (21%) and verification bias (19%), which occurs when only a proportion of the study group receives confirmation of the diagnosis by the reference standard. Most of the included articles in this systematic review have only a small portion of the sample tested for genetic validation. As regards the radiomics model development, the limitations were the lack of external validation (21%) and observer variability (15%), because of variation in radiomic procedures including, but not limited to image segmentation, features reduction and selection, model testing, etc., so a radiomic model may not consistently yield the same result when repeated and a lot of the studies did not perform any repeatability assessment. Additionally, the third most common bias was the high dimensionality or also called small n-to-p bias (14%) that occurs both in the radiomics feature and genomic analysis. Test technology bias (9%) and disease severity bias (7%) were reported in 10 and 8 studies, respectively. Test technology bias is a result of technological improvement due to time, meaning that estimates of test performance may be affected, while disease severity bias refers to differences in disease severity among studies that may lead to differences in estimates of test performance. Less commonly reported limitations or biases in the included studies are population bias (4.5%), selection bias (3.6%), and disease prevalence bias (1.8%), all related to how the sample or population was chosen for the study. The last two reported biases only occur in one study each. They are clinical review bias and arbitrary choice of the threshold value. These are biases that typically occur during the interpretation or the analysis of the results. The full list of limitations and biases evaluated for each study can be found in Appendix B. The summary of all included studies and their characteristics can be found in Table 1.

### 2.2. Brain Tumors (Gliomas/Astrocytomas)

A total of 15 studies addressed radiomic features in various subtypes of gliomas and examined correlations with a wide range of genomic data (see complete data in Appendix A). They all focused on survival prediction and almost all studies extracted their radiomic features from MRI imaging. Only one study used radiomic features extracted from FDG-PET [18]. All studies included a decent number of patients (85–435, mean 165). RQS varied in the studies (5–19, mean 12) but had a mean score similar to the included studies in this review.

Two studies assessed the value of radiomics extracted from MRI imaging to predict *MGMT-methylation status* and *IDH genotypes* in grade II-IV astrocytomas [15,20]. Wei et al. found that four single radiomics signatures (T1-WI-tumor, T2-FLAIR-tumor, T1-WI-oedema, T2-FLAIR-tumor) had significantly different expressions in the *MGMT* methylated and unmethylated groups. These findings were consistent in both the training and the validation cohort (AUC > 0.7). The study also concluded that ADC-tumor and ADC-oedema did not perform well to predict *MGMT-methylation*. The best prediction performance proved to be a combination of the four signatures (AUC > 0.925, 95%CI (0.861, 0.989)). The combined signatures also turned out to be a significant predictor of survival in high-risk and low-risk subgroups. In the study that assessed *IDH-genotypes* [20], it was found that a model comprising CE-T1WI + T2-FLAIR + ADC MRI sequences showed the best performance in predicting IDH for low- and high-risk groups in grade II and grade III-IV astrocytomas. The training cohort showed an AUC of 0.874 and 0.928, respectively, in the low- and high-risk group, and an AUC of 1.000 and 0.722 in the validation cohort (95%CI [0.886–0.916] in training, [0.845–0.930] in validation). The combined model of the three MRI sequences did also provide statistically significant discrimination between the high- and low-risk groups. These findings suggest that a combination of single signature radiomic profiles can provide a diagnostic and prognostic value in grade II-IV astrocytomas.

Four studies focused on the significance of *IDH mutation* and MRI-based radiomics in the survival prediction of patients with gliomas [13,16,21,45]. They all found that both the radiomic features and *IDH* are independent prognostic parameters that correlate with survival. One study determined that the most significant radiomic feature to predict survival was based on the heterogeneity in glioblastoma (GBM) [13], while another found the most relevant feature to be related to shape [21]. Qian et al. found a nomogram, constructed of radiomic, clinical and molecular risk factors (23 genes described in the article), to be the most efficient tool to estimate survival [16]. The last study found that several selected radiomic features could effectively divide gliomas into two survival-related subgroups, where the subgroup with superior survival and lower malignancy-rate, had a higher probability of *IDH-mutation* [22]. The findings of these studies, however different, show a correlation trend between IDH-mutation, radiomics, and survival outcome.

Two studies assessed the correlation between *MGMT-methylation*, radiomics, and survival in patients with gliomas and glioblastomas. The features were extracted from FDG-PET and MRI images [18,19]. The study using FDG-PET images found the radiomics model to be performing best in predicting methylation status in both the training and the validation group. The findings were compared to the clinical signature and the combined model of the radiomic and clinical signature [18]. The other study that extracted the features from MRI images, found limited use of radiomics to predict *MGMT-methylation* status, but found a good correlation between survival and radiomics [19]. One study supported the positive findings on survival prediction with *MGMT-methylation*, *IDH-mutations*, and radiomic features [24]. It showed that combining the two mutations in the prognostic model improved the performance.

Two studies examined the correlation between subgroups of either radiomic features or gene expression to predict survival [23,26]. One used a clustering method to identify imaging prognostic biomarkers, by dividing GBM patients into three groups depending on their MRI radiomic features [23]. Cluster 1 was characterized by heterogeneity in signal intensity and associated with lysosomal activity and autophagy; cluster 2 was characterized by shape/texture and was associated with chemotaxis and pro-inflammatory response. Cluster 3 was characterized by the complexity of the texture and showed decreased activity in the MAPK pathway, which is one of the frequently altered pathways in GBM. Cluster 3 exhibited the most favorable prognosis.

The other study divided lower-grade glioma (LGG) patients into subgroups according to their DNA copy-number subtypes (CN1–CN3). The expression of *IDH-mutations* and *1p19q codeletion* (combined loss of the short arm chromosome 1) were different in the three subgroups. CN1 was associated with hemorrhage, poorly defined margin and volume, CN2 with extranodular growth and width, and CN3 was associated with necrosis/cystic, hemorrhage, poorly defined margin, and frontal lobe tumors. CN2 turned out to be the subgroup associated with the shortest median PFS and OS. These findings and methods exhibit a correlation between the genomic signature and radiomics, whether the patients are grouped by tumor features or tumor gene expression [26].

Five studies had a more unique assessment of radiomics and genomics in glioma patients [12,14,17,24,25]. One concerning GBM patients used six metagenes (*WDR72, C14orf39, TIMP1, CHIT1, ROS1, EREG*) derived from a differentially expressed gene (DEG) analysis and four machine learning algorithms on radiomic features to examine a correlation with survival [14]. They found that the algorithm GBDT (gradient boosting decision tree) had the highest accuracy for predicting patients with under or over 1-year survival time. All six metagenes had significantly different expression levels in short versus long term survival patients, and three of the metagenes (*TIMP1, ROS1, EREG*) were positively correlated with nine image features, but the study does not state whether the radiomic algorithm is a significant predictor of the genomic status in patients. A study also concerning GBM patients identified 192 DEGs in two groups, low-risk and high-risk groups, obtained from a radiomic risk score (RRS) [17]. The analysis revealed that the low-risk group was associated with *BMP4* gene expression when compared to the high-risk group. TP53-inducible proteins and TP73-AS1 expression were also statistically different between the two groups. These two studies show a clear development in the genetic profile of radiomic studies and enlighten the complexity of the radiogenomic analysis.

Another study classified GBM patients by performing a pathway analysis based on gene expression profiles from 1740 different genes, and divided them into four subgroups; proneural, neural, classical, and mesenchymal. Based on MRI, the extracted features have the following characteristics; Necrosis (NE), Contrast Enhancement (CE), Edema (ED), Tumor Bulk (TB), and Total Tumor Volume (TV) [12]. The results showed that three features (NE, CE, and TB) performed best in predicting overall survival and assessing low and high-risk survival groups. ED and TV were the strongest predictors of subtypes, but the overall prediction of subgroups by features performed poorly. A similar study also used pathway analysis to classify subgroups [25] and extracted 30 key genes for genomic analysis. The genes are not specified in the article but are divided into red and blue modules. The red module is characterized by immunity, inflammation, and proliferation, and the blue model by cellular signaling pathways. They are both correlated with 13 prognostic MRI extracted radiomic features. The findings in these two studies show a more complex genomic correlation as multiple genes are implemented in the radiomic signature.

One study examined the genetic profile of a hypoxia-derived pathway [17]. The authors identified 21 genes that are implicated in the hypoxia pathway of GBM, for example, the *vascular endothelial growth factor (VEGFA) gene*, *angiopoietin-like 4 (ANGPTL4)*, and *galectin-3 (8LGALS3)*. The top eight radiomic features extracted from MRI imaging were found to be significantly associated with survival. They also found that high radiomic feature expressions corresponded to a high hypoxia enrichment score. Low radiomic feature expressions corresponded to a low hypoxia enrichment score. Since the extent of peritumoral hypoxia was related to prognosis, hypoxia monitoring could be a possible contributor to clinical practice by the evaluation of treatment response and resistance.

In summary, specific radiomic signatures show promising potential to become an important clinical tool for noninvasive diagnostics and prediction of prognosis in patients with gliomas.

### 2.3. Lung and Head and Neck Cancers

There were eight studies addressing lung, head, and neck cancer (see complete data in Appendix A). All of the studies used CT-derived radiomic features and two studies [32,34] also had PET FDG-derived radiomic features. Though there was a high proportion with a decent number of included patients (24–837, mean 310), most studies had few patients with genetic data and only one study had data on multiple genes for more than 100 patients [33]. The studies had a varied RQS (5–17, mean 12) with a mean score equal to the mean of the other studies in the review.

The studies choose many different ways to focus on lung cancer. In one study [31], only late-stage patients were included. Two studies [29,51] only included early-stage lung cancer patients. The rest of the lung cancer studies had a mix of patients dominated by early-stage patients.

The studies included different subtypes of lung cancer, and small cell lung cancer was typically excluded. Only two studies allowed small cell lung cancer [31,33], though one [33] only had 3% cases with this subtype. One study focused on adenocarcinomas [51] and one focused on squamous cell carcinomas [29]. The rest of the studies had a mixture of non-small cell lung carcinomas (NSCLC). Only one study [28] included head and neck cancer patients, and it also included lung cancer patients.

The handling of genetic data in the studies on lung cancer can be divided into three subgroups. Two studies [28,32] focused solely on the *EGFR mutation*, three studies [31,33,34] looked for correlations with multiple single genes and three studies [28,29,51] focused on correlation to gene pathway alteration.

The only study with head and neck cancer (H&N) data [27], was the oldest in the review and was published in 2014, one of the first-ever studies published on radiogenomics. The authors generated a four feature radiomic signature on a lung cancer data set and tested it on two H&N cohorts. They found a high and similar performance in H&N1 and H&N2 (both CI = 0.69, *p* < 0.001) and better performance than in their lung validation data set (CI = 0.65, *p* < 0.001). They found that 15% of the radiomic features derived from their lung cancer training set were significant for survival in all three validation data sets. Unfortunately, they only had genetic data on NSCLC patients.

Most studies chose to look at overall survival, disease-free survival or both, and they all found significant associations between their radiomic models and survival. In one study [32] these continuous data were handled in a dichotomous way and it was found that hyper progression (time-to-treatment failure (TTF) < 2 months) and durable clinical benefit (DCB, PFS ≥ 6 months) were associated with their model. Kirienko et al. [34] tried predicting cancer histotypes and relapse. Aerts et al. [28] tried to predict response to the EGFR inhibitor Gefitinib treatment and found that no features were predictive on post-treatment scan (*p* > 0.08), but the change in features between the pre-and post-treatment scans was strongly predictive (significant feature AUC-range = 0.74–0.91,) for response to Gefitinib. This proves that radiomic features can be a possible supplement in discriminating the true response from cancer pseudoresponse.

One of the studies focusing on the *EGFR gene* [28] found that the Laws energy feature was significantly predictive of EGFR-mutation status (AUC = 0.67, *p* = 0.03). Laws energy emphasizes edge, spot, ripple, and wave patterns. In the other EGFR gene study, [32] made a deep learning model (convolutional neural network) and hence did not extract single radiomic features. The authors found that their EGFR-mutation prediction model yielded AUC of 0.83, and 0.81 in the validation, and external test cohorts, respectively.

In the group of studies looking at multiple genes, the smallest study only had 24 patients [31]. They found the *TP53 gene* to be significantly correlated to their high-risk patient cluster and showed that the amount of free circulating TP53 DNA two weeks after chemotherapy was correlated to their model. Both EGFR and IDH1 mutations were only found in their high-risk cluster. Perez-Morales et al. [33] found six genes associated with their model, three of which were uncharacterized. The other three were *FOXF2, TBX4*, and *TM4SF1*, which are all involved in cell proliferation or cell growth. The third study looking at multiple genes [34] was different from the others in that the authors did not use gene data to validate their radiomics model but developed different models for radiomics, genes, and radiogenomics. They analyzed 238 genes. Their combined radiogenomic model for predicting histotype (squamous cell carcinoma or adenoma) contained the same two genes (*HIF1A* and *TP63*) as their best model based on genes alone and no radiomic features. Their best performing model in predicting relapse (yes/no) built on the genes *CXXC4*, *PAK3*, and *GHR gene* and the LRLGE (low/high gray-level emphasis) radiomic feature; it had an AUC of 0.87.

The studies using gene set enrichment analysis found correlations between radiomic models and biological functions that intuitively are associated with survival in cancer patients. One study [27] found that two of the four studied radiomic features are measurements of intratumor heterogeneity, and they strongly correlated with cell cycling pathways, indicating an increased proliferation of more heterogeneous tumors. Another study found that the authors’ radiomic model was associated with metabolic processes and the immune system [30], and the third study found a correlation between apoptosis and proliferation-related genetic pathways [29].

### 2.4. Breast, Ovarian, and Endometrial Cancer

In the present review, three studies were focused on breast cancer, three on ovarian cancer, and one on endometrial cancer (see complete data in Appendix A). Breast cancer studies were all based on radiomic features extracted from MR images, two of which analyzed dynamic contrast-enhanced MR images [35,36]. The gene expression profiles were assessed through genetic databases, i.e., the TCGA or/and the GSE for partial or total data collection in two of the studies [35,36]. The study population varied (*n* = 84–1362, mean 586), but with a generally high proportion of patients with genetic data. The mean RQS was higher than the mean RQS across all studies included in the review (10–16, mean 14).

The clinical outcome ranged from the risk of recurrence [35], prognosis/survival [36], and pathological response after therapy [37]. The genetic data also varied and were based on gene pathway analysis in one study [36], multigene analysis in another [35,37], and lastly single-gene analysis in the third study focusing solely on the *human epidermal growth factor receptor 2 (HER2) gene* and its protein [37]. As for limitations and risk of biases, two of the three included studies did not receive external validation [35,37], while one study suffered from detection and verification bias [36].

To assess the risk of cancer recurrence, Li et al. investigated the correlation between MR computer-extracted image phenotypes, including tumor size and enhancement texture patterns, and the risk of breast cancer recurrence using gene expression assays of MammaPrint, Oncotype DX, and PAM50 which showed significant associations (R2 = 0.25–0.32, r = 0.50–0.56, *p* < 0.0001) [35]. Wu et al. focused on finding signaling pathways associated with prognostic tumor-adjacent parenchymal imaging features from MR imaging. They found that tumor-adjacent parenchymal imaging features were associated with tumor necrosis signaling pathway and poor breast cancer survival [46]. Lastly, the prediction of the pathologic response to neoadjuvant chemotherapy was investigated by Bitencourt et al. They demonstrated, with a diagnostic accuracy of 83.9% (52/62), that MRI-based clinical and radiomic features coupled with machine learning were able to predict pathologic response after neoadjuvant chemotherapy in HER2 overexpressing breast cancer patients [37]. These studies shed valuable light on the ability of radiomic and radiogenomics models of breast cancer to predict clinical outcomes and potential selection of treatment strategies.

Three studies addressed ovarian cancer and the radiomic features were all extracted from CT images. Only one of the ovarian studies assessed partial data collection of gene expression profiles through genetic databases [40]. The study population varied amongst the studies (38–364, mean 163) but all included patients who received genetic validation. The mean RQS of studies on ovarian cancer was lower than the mean RQS across all included studies and varied considerably (5–20, mean 11).

Survival (progression-free survival/overall survival) was the primary clinical outcome of focus in all of the included ovarian cancer studies. The genetic data were based on gene pathway analysis in one study [40], whereas the remaining two studies focused on single gene analysis, that is either *BRCA mutation* status [39] or Classification of Ovarian Cancer transcriptomic (CLOVAR) subtypes (differentiated, immunoreactive, mesenchymal, and proliferative), and amplification of *19q12* involving *cyclin E1 gene (CCNE1)* [38].

Lu et al. built a radiomics-determined mathematical descriptor from CT images of epithelial ovarian cancer that consisted of four radiomics features and demonstrated that it can be utilized to predict survival and associated with a stromal phenotype and DNA-damage response pathways [40]. The study suffered from possible selection bias, and further investigation into heterogeneity for bilateral ovarian tumors is needed. The remaining two studies focused on inter-site texture heterogeneity between/across different metastatic sites in patients with high-grade serous ovarian cancer and reported associations between inter-site heterogeneity texture metrics and clinical outcomes including survival [38,39]. Furthermore, Vargas et al. found that amplification of *CCNE1* predominantly occurred in patients with more heterogeneous inter-site textures [38], whereas the study conducted by Meier et al. did not find significant associations between texture metrics and *BRCA mutation* status [39]. Neither of studies received either internal or external validation.

There was only one included study focusing on endometrial cancer with an RQS of 12 and MRI-derived radiomics features from patients with endometrial cancer [41]. The study population was large (*n* = 487) but only a small portion had full genetic data (*n* = 51) and was partly assessed through the TCGA database. Hoivik et al. generated radiomic clusters composed of patient groups with differential risk profiles and linked these radiomic clusters to differential gene expression and demonstrated that whole-volume tumor radiomic profiling from manual tumor segmentation could identify patients with high-risk histological features and poor survival. In addition, they reproduced the same radiomic prognostic groups with radiomic profiling by automated tumor segmentation. Furthermore, an 11-gene high-risk signature was defined and associated with poor survival (copy-number-high/p53-altered). The study suffered from selection, test technology and verification bias, as well as a lack of external validation.

### 2.5. Urogenital Cancers (Kidney, Bladder, and Prostate)

There were six studies on renal cancer, one study investigating bladder cancer, and one on prostate cancer (see complete data in Appendix A). The six studies of clear cell renal cell carcinoma (ccRCC) were all based on radiomic features extracted from CT images and all of them made use of the TCGA- and/or TCIA-database for partial or total data collection. Concerning all included articles in this review, the ccRCC studies generally presented large study populations (214–520, mean 373) and low to moderate RQS (6–15, mean 12). Five out of six studies had external validation. The most frequent limitation was verification bias (4 of 6 studies) but otherwise, the limitations of the six studies were various.

All of the ccRCC studies showed significant results regarding the radiomic and radiogenomics models’ ability to predict outcomes in the form of PFI [44,45], MFS [42], and OS [42,43,46,47]. Zeng et al. integrated radiomics with genomics, transcriptomics, and proteomics into a multi-omics model which was significantly better in terms of predicting 1-year, 3-year, and 5-year OS compared to models formed by single omics and the radiogenomics model, which shows potential, but, as the authors outline, the correlation between different omics features is complex and needs to be investigated further [47].

The genetic validation in the ccRCC studies ranged from gene expressions found correlated to the radiomic features [42,44,45,46,47] and hypoxia-related genes associated with survival [43]. Several of the studies showed associations between radiomics features and molecular functions in the form of mRNA expression [46,47] and biological pathways such as T cell activation [44] and proteasome, cell cycle, and p53 signaling pathway genes [45]. Regulatory T cells play an essential role in the progression of ccRCC in internal and peripheral tissues [57,58]. Furthermore, several drug–target relationships between radiomic feature-associated gene expression and tumor response were identified in pT1 ccRCC-patients with postoperative metastasis [42], which could pave the way for radiogenomics models to predict or monitor response to immunotherapeutics in patients with ccRCC.

The few studies of BLCA [48] and prostate cancer [49], and the mutual limitations of the included articles in the form of small study populations (62 and 64 patients each), no external validation, and low RQS score (8 and 10), can indicate that the research in radiomics regarding outcome assessment with genetic validation in these types of cancer is still in the commencement phase.

The study by Lin et al. [48] was the only one investigating urothelial bladder carcinoma (BLCA) included in the review. It was found that the CECT radiomics signature was a highly significant prognosticator for PFI, and gene functional enrichment analysis showed that angiogenesis was the most significant radiomics signature-associated biological process. Angiogenesis plays a significant role in the development of cancer owing to the need for an adequate supply of sustenance and removal of metabolic waste to and from the tumor site [59], and angiogenesis has been utilized as a therapeutic target in urothelial carcinoma [60]. Another study dealing with BLCA [61] was excluded due to incomplete information regarding the type of radiologic modality used for feature extraction, but they presented seven signature radiogenomics models to reveal the m6a methylation status which was correlated to the outcome for BLCA patients.

The only included study focusing on prostate cancer was Hectors et al. [49], which showed a significant correlation between MRI radiomic features and gene signatures. The Post-Operative Radiation Therapy Outcomes Score (PORTOS) signature had the strongest correlation, and this signature has previously been demonstrated to be associated with the risk of the development of distant metastasis [62]. Only one other article about prostate cancer [63] came up in our database search, but it was excluded due to a lack of clinical outcome assessment.

### 2.6. Gastrointestinal Tumors (Esophagus, Gastric, Colon Cancer, and Hepatocellular Carcinoma)

In the present review, six articles in total dealt with gastrointestinal tumors, two studies were focused on esophagus cancer, one study dealt with gastric cancer, one on colorectal cancer, and lastly one on hepatocellular carcinoma (see complete data in Appendix A). The two studies that dealt with esophagus cancer were based on radiomic features extracted from CT images and focused on gene expression for the pathophysiological explanation of the models. Both articles included a good number of included patients (106–231, mean 168), but had small samples for the genetic analysis (28–40, mean 34). Both studies had very high RQS (both 17) in comparison to the mean of included studies in the review, and although both had external validation, they still had high dimensionality bias, also known as small n-to-p bias, especially in the genetic data.

Both studies focused on the use of radiomic models to predict treatment response in patients with esophageal squamous cell carcinoma after neoadjuvant chemotherapy. Hu et al. [50] developed a model integrating intratumoral and peritumoral radiomics features that achieved improvement in predictive performance (AUC 0.852, 95%CI [0.753–0.951]) compared with the conventionally constructed model merely using intratumoral radiomics features (AUC 0.730, 95%CI (0.609–0.850)) to estimate pathological complete response. Gene sets associated with the combined model mainly involved lymphocyte-mediated immunity. The association of the peritumoral area with response identification might be partially attributed to type I interferon-related biological processes and this underlined the potentially important role of the surrounding stroma or peritumoral tissue in therapy resistance. The study shows that peritumoral radiomics may provide an additional predictive value for treatment response estimation in esophageal squamous cell carcinoma and it underlines its significance in assessing treatment response in clinical practice [50].

The study by Xie et al. included a two-time-point delta radiomics analysis in a prognostic prediction model in conjunction with a biological underpinning feature selection method. One delta radiomic feature (correlation from GLCM families) was significantly more predictive of patients’ survival than the study’s changes in tumor volume. This is similar to the outcome of previous studies, which have shown that change in radiomic features could outperform the volume measures for disease evaluation [28,64,65,66]. The study differs from other studies included in the review, in that the genomic association was a useful method for the radiomic feature selection. The authors, instead of analyzing the underlying biological process by gene enrichment analysis, used these genomics data as a new feature selection filter to assure biological robustness [51].

The study addressing gastric cancer provided evidence that a prognostic radiomics signature could capture tumor cell intratumor heterogeneity, which was also associated with underlying gene expression patterns. The authors built a radiomics signature for predicting the overall survival (AUC 0.803) and disease-free survival (AUC 0.753) of patients with complete resection of their gastric tumor to further identify candidate benefits from adjuvant chemotherapy (*p* < 0.001). The radiomics trait-associated genes captured clinically relevant molecular pathways and potential chemotherapeutic drug metabolism mechanisms, including chemokine regulation [31].

The study by Negreros-Osuna et al. explores the potential of radiomics texture features to enable the identification of the presence of *BRAF mutation* and the prediction of 5-year OS advance stage (IV) colorectal cancer (CRC). They found that CRC tumors with *BRAF mutation* show lower values of the derived radiomics texture features standard deviation and mean value of positive pixels of the tumor region of interest on CT images in comparison with *wild-type BRAF*. This also showed that the CRC tumors with less radiomics texture heterogeneity behave more aggressively than those showing more heterogeneity, and this was associated with unfavorable 5-year OS survival [53]. This provides evidence that radiomics can potentially differentiate *wild-type BRAF* CRC tumors from those with *BRAF mutation* and predict OS in advanced-stage disease.

Hectors et al. [54] was the only included study focusing on hepatocellular carcinoma (HCC), and it showed a correlation between MRI radiomic features and PD1 and CTLA4 at mRNA expression level, and the significant predictive ability of the radiomic features regarding early HCC recurrence, but it did have some limitations and a moderate risk of bias. The study population was small (48 patients), there was no internal or external validation, the RQS was low (7), and it displayed small n-to-p bias. The study did not produce a radiomics model because several of the MRI scans were inadequate.

### 2.7. Others (Melanoma, Solid Tumors)

In this last section, we present studies that did not fit with any of the previous categories but were still included due to the interesting results. These results related both to molecular oncology and the possible applications of radiomics biomarkers in cancer patients, as both focused on CD8 cell expression (see complete data in Appendix A).

The first study assessed prognostic biomarkers in 52 patients with late-stage melanoma (III/IV) that underwent pre-treatment PET-CT scans. Using a non-invasive tumor assessment, widely available in radiology practices, we found a prognostic signature derived from PET-CT imaging that represented a “hot” tumor immune microenvironment. They showed that CT parameters, high mean of positive pixels (MPP4), and high standard deviation (SD3), were associated with patient survival. These biomarkers appeared to have better prognostic value than primary tumor ulceration status, which forms part of the staging system. They also identified a correlation between the radiomics parameters and tumor biology (*BRAF status*, immune signatures, and CD8 expression) [55]. As late-stage patients often have multiple tumors in difficult access sites, this technique could allow patients with the unresected disease to be treated with a precision medicine approach as radiomics parameters are derived from routine images, including BRAF inhibitor and/or other immunotherapy treatments for melanoma.

Sun et al. [56] presented a CT radiomics-based biomarker of tumor-infiltrating CD 8 cells in patients with solid tumors treated with anti-PD-1 and PD-L1 immunotherapy. The solid tumors included a long list of different cancer types in various organs (lung, gynecological, breast, head and neck, etc.), which caused a heterogeneous study population, but the authors prevented overfitting to a particular subset of the tumors by selecting statistically different cohorts for training and testing of the biomarker. The biomarker was successful in predicting the gene expression signature of CD8 cells (AUC 0.67, 95%CI (0.57–0.77), *p* = 0.0019), response to anti-PD1 and PD-L1 treatment, and overall survival.

## 3. Discussion

There is enough evidence to state that new science has emerged, a novel research field focusing on establishing an association between radiological features and genomic or molecular expression to shed light on the underlying disease mechanism and enhance prognostic, risk assessment, and treatment response models toward personalized medicine [11]. The oldest study included only dates to 2014 [27] and shows that there has been exponential growth in the last couple of years. Fifty percent of our included studies were published between 2020 and 2021.

On their own, these studies are very unique; they use two big data sciences, radiomics, and genomics, to pursue clinical outcomes (Figure 8). The study design of these studies is very complex and currently does not fit with any of the known types of studies and could become a new category by itself. We have represented it graphically as a three-side study or a triangle. The normal radiomics workflow, which starts with the acquisition of raw images and continues with some form of pre-processing of the information, extraction, and selection of features, and developing models for a particular outcome is one side. The second side includes finding or discovering a relation between the image-extracted features against a genomic variable that can be exploratory or hypothesis-driven. The third side of the study validates or determines a biological explanation between the radiomic models and the pre-determined outcome. To accomplish such studies, multidisciplinary coordination is required between the different groups that work on them. It requires a team where oncologists, radiologists, radiographers, nurses, geneticists, molecular biologists, etc., but also statisticians, data analysts, and medical engineers carry off the different tasks demanded by the studies.

Convincing evidence has emerged showing that there is an association between different radiomic models and genomic data from various tumor types to assess different outcomes (Table 1, Appendix A), bringing biological meaning and genetic validation to imaging features. Several genetic sciences were applied to the studies with transcriptomics playing a doming role, and the cancer types with most radiogenomics studies were glioblastoma, lung cancer, and renal cell carcinoma. This is interesting in the sense that these cancer types are not the most common tumors out there, but recent research on oncogenesis, including cell-of-origin patterns, oncogenic processes and signaling pathways, is probably fueling possible premises and allowing hypothesis-driven research in these specific tumors.

We also need to point out that there are two major factors facilitating the development of these studies. One is the use of artificial intelligence (AI) through both machine learning and deep learning at different steps of the radiogenomics process. The second one is the availability of both genetic and imaging databases. The Cancer Genome Atlas (TCGA), a landmark cancer genomics program, allows researchers to access information that otherwise would be impossible to access. It has helped to establish the importance of cancer genomics, transformed our understanding of cancer, and even begun to change how the disease is treated in the clinic. The impact goes even further as it extends to other research fields, such as radiogenomics [9].

We found many of the same limitations as reported and established by other authors [2,11,67]. One of the major challenges is the lack of reproducibility and generalizability of the reported radiomics signatures (features and models). Sources of variation exist in each step of the workflow; some are controllable or can be controlled to certain degrees, while others are uncontrollable or even unknown [66]. These are very well-known limitations of radiomics, and genomics not only brings the same problems to radiomics studies but also increases the high dimensionality and small n-to-p data bias. Although this was an important bias in our limitations list, it was not listed often in the reviewed articles. This is because the studies applied the technique to limit the effect of multiple testing, feature reduction, and gene selection this bias was not recorded. Therefore, even if it is underreported in our review, it is still a big concern, especially because one of the running hypotheses is that biological validation could reduce these limitations [6].

Despite the rapidly increasing body of publications, there is no real clinical use of a developed radiomics signature so far. This is not surprising when one realizes that there is a low level of evidence throughout the entire field and all the included studies are categorized as level 4 of evidence, even though all included studies had good methodologies, robust statistics, and thorough result reporting (Oxford Centre for Evidence-Based medicine 2011 Level of Evidence, http://www.cebm.net/index.aspx?o=5653, accessed on 2 May 2022). The difference in their quality was most evident in the type of analysis that was conducted, and we classified them broadly into exploratory versus hypothesis-driven studies. Most of the included studies that were exploratory in nature, often feasibility studies and very few had a clear objective or were hypothesis-driven studies. None of the included studies were prospective in nature, and none used a previously developed radiogenomics or radiomics model.

Insufficient transparency in reporting radiomics studies further prevents translation of the developed radiomics signatures from the laboratory into the clinic. Our research team was stunned by the amount of underreported data or missing data. A lot of the information needed to complete the risk of bias analysis was difficult to find; some information was found in Appendix A or previously published studies, but some data was never found, and it is stated as such in the result tables. Editors and peer-review journals will need to adapt to this type of research, as massive amounts of data are used and many steps are undertaken before a radiomics model is developed and analyzed with their corresponding genetics data. Missing data hinders the process and obstructs the possible translation of these studies into the clinical setting and the capacity to be evaluated properly making reproducibility a bigger issue.

Because of the large variability in the radiogenomics process, reading and understanding these studies is challenging and this highlights the need for guidelines and tools that facilitate the integration of these sciences [9]. New technology can improve current problems and surely the use of AI and deep learning will keep expanding. Currently, the radiomic quality score (RQS) is the only known tool of this kind for radiomics studies, to the knowledge of the authors. We used it as a standardized method to evaluate the different steps taken to develop the radiogenomic signatures and models, as it also takes into account that non-radiomic variables (for example, BRAF mutation) are used in the multivariable models to provide a more holistic model. It also considers that biological correlates are used in the study to demonstrate phenotypic differences or deepened the biological understanding of the radiomic models. It is often criticized because of the ways it categorizes and scores the studies, even though it is often used to give guidance, as we have used in our study. The total score is 36 points; no study came near that; the highest score was 20 in our review. In addition, many criteria from the tool are unrealistic to achieve in one study at a time, as many could be a study by itself or implicate multiple hypotheses. There is no doubt that there is a need for such tools, but the current one needs to adapt better to what is possible and be pragmatic as well.

Ultimately, it is important to comment on future directions for this new science, commenting on exciting results from different studies from both included and excluded articles from the review. Another limiting factor is the number of patients that have tissue samples available and genetic data extracted from them; it is not only expensive but requires high levels of expertise. This was a limitation in almost 20% of the studies, where not all patients received genetic validation. A study by Latafa et al. [31] solved this problem by performing a liquid biopsy of cell-free DNA (cfDNA). Not only did they extract the data prospectively, but they also extracted it at several time points to test the response to treatment in patients with advanced lung cancer. A study that focused on the development of radiomic signatures from perivascular adipose tissue to improve cardiac risk predictions utilized adipose tissue biopsies from patients undergoing cardiac surgery [68]. This study is exciting as a creative way to obtain tissue samples and illustrates how radiogenomics can be applied to other diseases other than cancer. There are examples of the application of radiomics methodology to cardiovascular disease [69], and in the aging population [70], so it is a matter of a short time before radiogenomics moves into other diseases.

Traditionally, cancers are grouped and treated by site or organ system. One study grouped patients by a common alteration and not by the type of cancer [56]. We believe this is an interesting study design, as a common mutation allows for a common therapy target across various cancers. Target therapy matched to underlying alterations improves outcomes [71,72]. In the era of precision medicine to tailor therapies, predicting tumor genomics using noninvasive techniques is the desired aim, especially in patients with the advanced-stage disease, who may benefit from a specific molecularly targeted agent or a combination of such agents [53].

## 4. Materials and Methods

### 4.1. Search Strategy

This review followed the methods described in a published protocol in the PROSPERO register (ID 310058). To identify relevant articles the PubMed, and EMBASE electronic databases were searched using relevant search terms which included radiomics and genetics. The syntax used to search the databases is detailed in Table 2. Publications were included until 31 January 2022. Filters for Humans and English were applied.

### 4.2. Inclusion and Exclusion Criteria

Studies were included if: (1) they were radiomics studies with complex feature extraction performed in any imaging technique including MRI, CT, US, PET-CT but not limited to; (2) relevant clinical outcome was assessed including survival, treatment response, prognosis or risk-stratification; (3) biological validation of the clinical outcomes and/or radiomic phenotype with genetic data; and (4) the study was performed in cancer patients or a specific tumor. To achieve a more homogenous group of articles, studies that have used both the radiomic and genetic data to improve predictive models were excluded. Studies were also excluded if they were case reports, conference abstracts, conference proceedings, reviews/systemic reviews, study protocols, editorials, and other non-original research articles. Lastly, short communications were also excluded because they do not provide sufficient information to assess the radiogenomics methodological quality.

### 4.3. Selection Process

After the initial search in the electronic databases performed by two authors (FB and KH), the abstracts and metadata were transferred to rayyan.ai, a systematic reviewer software (https://www.rayyan.ai/, accessed on 28 March 2022). After eliminating duplicates, five reviewers (AA, FB, LH, KH, and SR) screened titles and abstracts to determine eligibility. Screening of the full text of publications was performed if the abstracts provided insufficient information to judge eligibility. All included studies were approved by at least two authors and uncertainties were resolved in consensus.

### 4.4. Data Extraction and Quality Assessment

The full text of included studies was divided and read among all five reviewers, who examined and analyzed the articles for risk bias and their quality. Qualified articles were read in full and relevant data was extracted, including first author, year and type of publication, country, funding source, imaging modality, disease/type of cancer, sample size, outcomes measured, genetic validation, and statistics used.

Given that the focus of the systemic review is on Radiomic Studies, the Radiomic Quality Scores (RQS) tool (https://www.radiomics.world/rqs, accessed on 25 April 2022) was used to assess the quality of the radiomics study design first. Studies with low or questionable methodology were excluded. The risk of bias was then assessed according to the Cochrane recommendation for diagnostic accuracy study Quality Assessment 2 (QUADAS-2). The QUADAS-2 tool was used to evaluate the diagnostic and/or predictive nature of the radiomics studies. The difference is that the reference standard questions of the QUADAS-2 tool were applied to both the radiomics and genetic parts of the articles being evaluated in the systematic review. Each reviewer applied tools independently.

The data of high-quality literature that meet the standard was summarized and analyzed and evaluated for limitations and biases that affected each study. The full list of limitations and biases can be found in Appendix B. Lastly, we evaluated whether the studies were exploratory research (sometimes called hypothesis-generating research) or confirmatory research (also called hypothesis-testing).

## 5. Conclusions

There is no doubt that radiogenomics is an exciting and promising new field of research. With the advance of next-generation sequencing techniques and machine learning algorithms, there are increasing high-throughput omics data. While genomics and radiomics have been studied individually, the integration of genomic and radiomic data into multi-omics-based machine learning models could provide new scope for precision oncology, which would aid a more comprehensive understanding and management of cancer diseases [51]. It takes around 15 to 20 years to transfer an innovation to the clinic, so it is not a surprise that radiogenomics has no clinical applications yet. To be able to translate this research into clinical development researchers in the field need to start asking themselves how to build knowledge to support its use beyond the outcome’s potential and feasibility. There is a need to determine the fidelity, acceptability, sustainability, costs, efficiency, and effectiveness of such technology, and it is wrong to think that if we develop it, clinicians will use it.

## Figures and Tables

**Figure 1 ijms-23-06504-f001:**
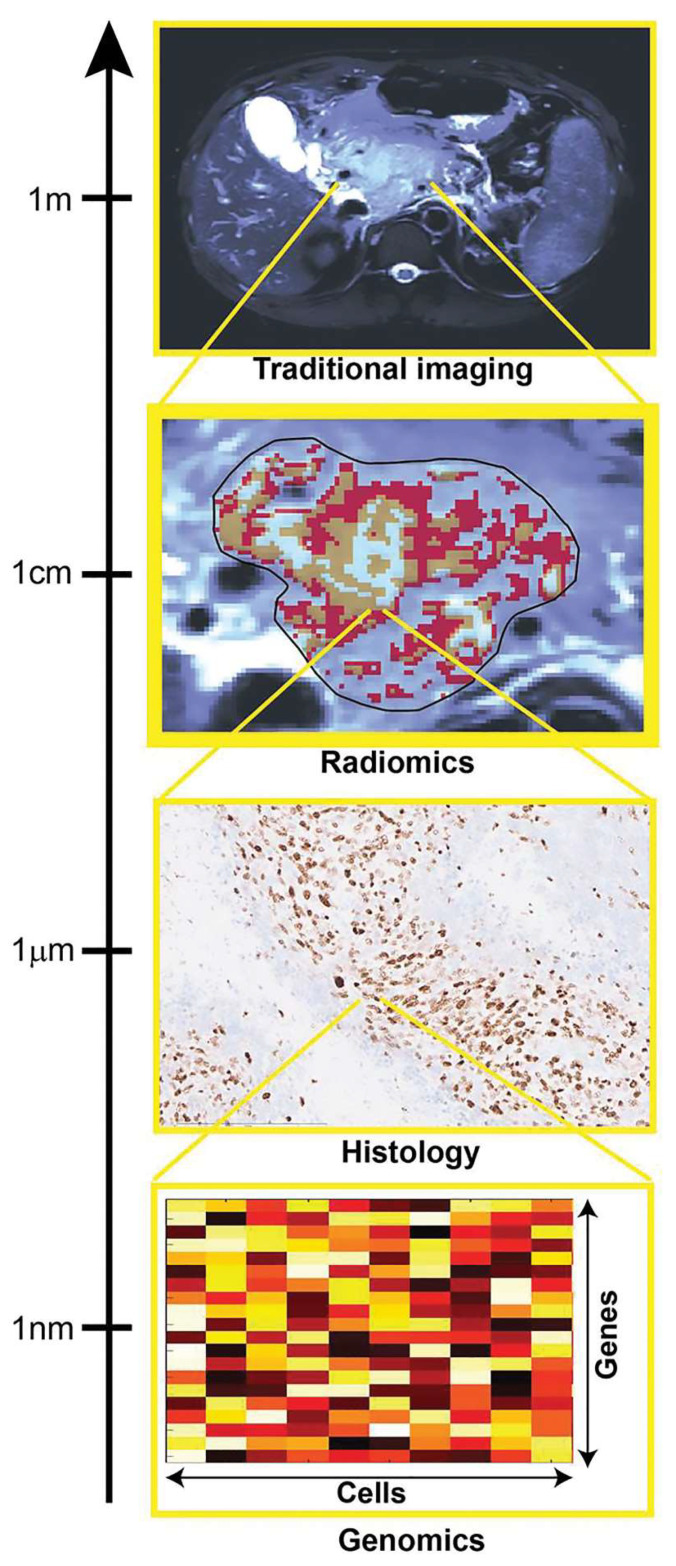
The image shows how multiscale quantification provides complimentary tumor insight. Histologic and genomic analysis can provide specific small-scale insight, which is useful for the validation of radiomic results, focused on quantification of spatial patterns of size exceeding image resolution [6]. Reprinted with permission of RSNA.

**Figure 2 ijms-23-06504-f002:**
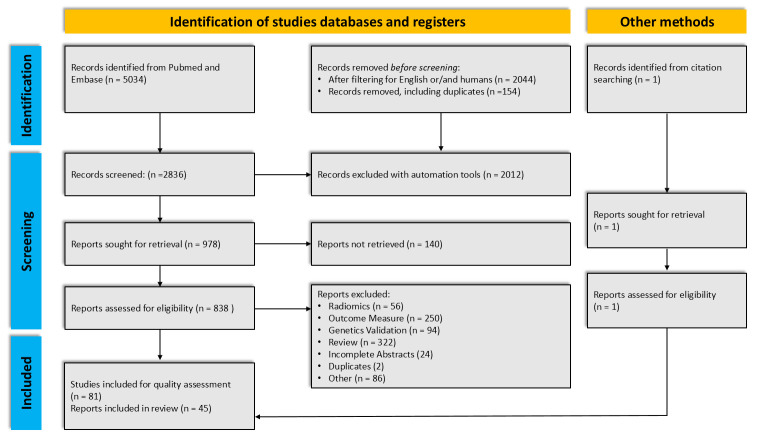
Flow diagram for the systemic review, which included searches of databases and other sources.

**Figure 3 ijms-23-06504-f003:**
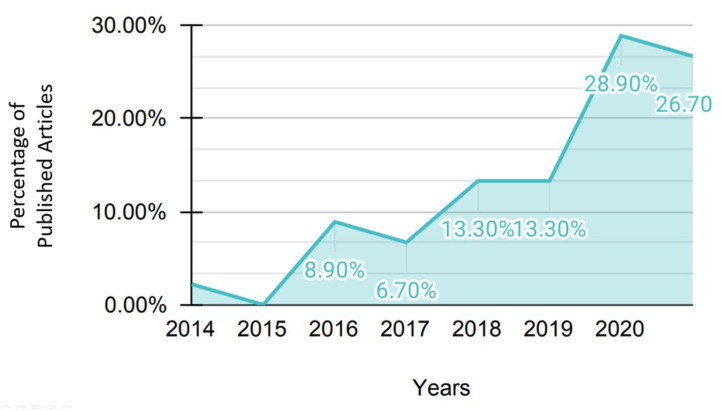
Stacked area chart of the percentage of published articles included in the systemic review by year of publication.

**Figure 4 ijms-23-06504-f004:**
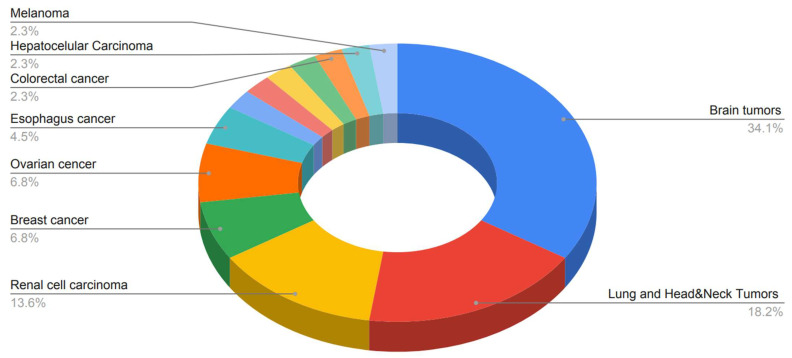
A pie chart of the different tumors studied in the articles included in the systemic review.

**Figure 5 ijms-23-06504-f005:**
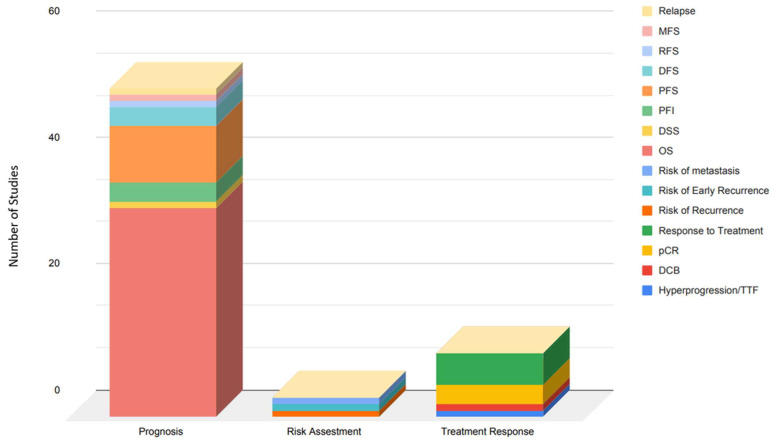
Stacked column chart of the number of studies grouped by the utility of the radiomics model and the outcome measure of the included articles in the systemic review. Abbreviations: MFS: Metastasis-free survival, RFS: Recurrence-free survival, DFS: Disease-free survival, PFS: Progression-free survival, PFI: progression-free interval, DSS: Disease-specific survival, OS: Overall Survival, pCR: pathological complete response, DCB: Durable clinical benefit.

**Figure 6 ijms-23-06504-f006:**
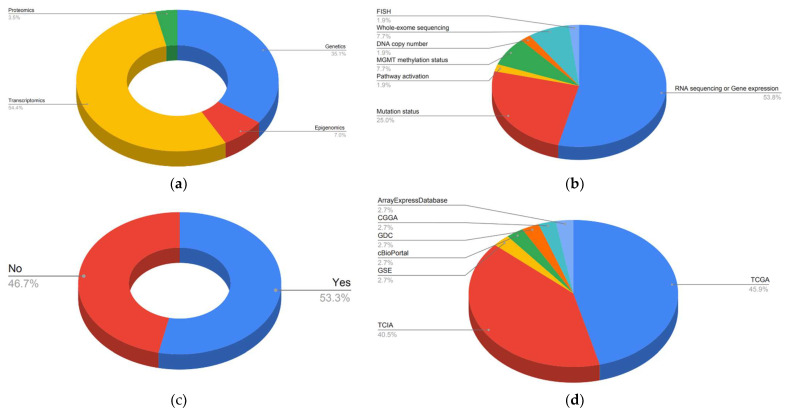
Pie charts of genetics data of the included articles in the systemic review: (**a**) Type of genetic science applied. (**b**) Analysis of the genetic data. (**c**) Used public genetics databases. (**d**) Public genetics databases used in the included articles.

**Figure 7 ijms-23-06504-f007:**
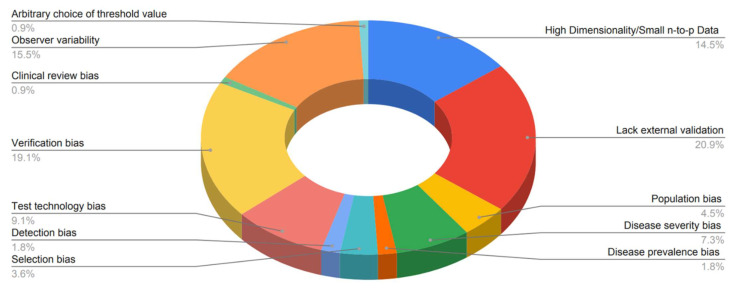
Pie chart of the limitations and bias of the reviewed publications.

**Figure 8 ijms-23-06504-f008:**
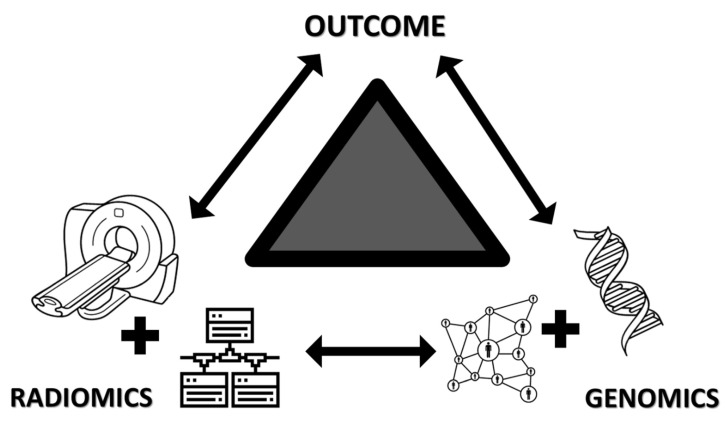
Graphical representation of summarizing the main characteristics of the included articles of the systemic review. Icons utilized in this figure were obtained from the Noun Project from the following authors: Oleksandr Panasovskyi (tomography), Pedro Baños Cancer (DNA), Gilbert Bages (network circles), Eucalyp (network boxes).

**Table 1 ijms-23-06504-t001:** Radiogenomics in practice—Studies Characteristics.

Utility/ Outcome Measured	Imaging Modality	RQS	Genetic Data	Genetics Database Used/ Name	Patients: Genetic Validation/ Total	Limitations **	Year/ [Ref.]
** *Brain tumors: Gliomas/Astrocytomas* **							
Prognosis/OS	MRI	8	GBM gene expression (mRNA)/Transcriptomics	Yes/TCGA and TCIA	91/141	12	2016 [12]
Prognosis/OS	MRI	6	Pathway activation/Transcriptomics and Proteomics	Yes/TCGA, TCIA, and cBioPortal	85/85	1, 2	2017 [13]
Prognosis/OS	MRI	13	Gene expression/Transcriptomics	Yes/TCIA and GDC	46/137	1, 12	2018 [14]
Prognosis/OS	MRI	13	MGMT methylation/Epigenomics	No	105/105	2, 6	2018 [15]
Prognosis/OS	MRI	11	Gene expression/Transcriptomics	Yes/TCGA and CGGA	85/85	1	2018 [16]
Prognosis/OS	MRI	18	RNA sequencing/Transcriptomics	Yes/TCIA	115/115	2	2018 [17]
Prognosis/OS	FDG PET-CT	16	MGMT methylation/Epigenetics	No	107/107	2, 7	2019 [18]
Prognosis/OS	MRI	13	MGMT methylation/Epigenetics	No	201/201	2	2019 [19]
Prognosis/ OS	MRI	13	IDH1 (2 mutations)/Genetics	No	105/105	2	2019 [20]
Prognosis/ OS	MRI	15	RNA sequencing/Transcriptomics	No	144/144	1, 2	2020 [21]
Prognosis/ OS	MRI	11	MGMT methylation, RNA sequencing/Epigenetics and Transcriptomics	Yes/TCGA and TCIA	85/166	1, 2, 10, 12	2020 [22]
Prognosis/ OS	MRI	19	IDH mutation status/Genetics	Yes/TCGA and TCIA	296/296	10, 15	2020 [23]
Prognosis/ PFS	MRI	17	IDH mutation, RNA sequencing/Genetics and Transcriptomics	Yes/TCIA	125/203	1, 12	2020 [24]
Prognosis/ OS	MRI	18	RNA sequencing/Transcriptomics	Yes/TCGA and TCIA	173/435	12	2021 [25]
Prognosis/ PFS and OS	MRI	5	DNA copy-number subtypes (CN1, CN2 or CN3)/Genetics	Yes/TCGA	153/153	1, 2, 6, 10	2021 [26]
** *Lung and Head and Neck Cancers* **							
Prognosis/OS	CT	16	Gene expression/Transcriptomics	No	89/1019	1, 3, 12	2014 [27]
Treatment Response	CT	13	EGFR sensitizing mutations/Genetics	No	49/49	1	2016 [28]
Prognosis/ DFS and OS	CT	8	Whole exome sequencing/Genetics and Transcriptomics	No	57/57	2, 3, 5, 10	2018 [29]
Prognosis/DFS	CT	13	RNA sequencing/Transcriptomics	Yes/TCIA	79/554	2, 12	2020 [30]
Prognosis/OS	CT	14	ctDNA TP53 mutations/Genetics	No	24/24	1, 2	2020 [31]
Prognosis + Treatment Response/ PFS, TTF, and DCB	FDG PET-CT	17	EGFR mutation/Genetics	No	681/837	5, 12	2020 [32]
Prognosis/ PFS and OS	CT	11	Gene expression/Transcriptomics	No	103/399	12	2020 [33]
Prognosis/Relapse + Histotype	FDG PET-CT	5	RNA sequencing/Transcriptomics	No	74/151	2, 12	2021 [34]
** *Breast Cancer* **							
Risk Assessment/ Risk of recurrence	MRI	16	RNA sequencing/Transcriptomics	Yes/TCGA	84/84	2, 10	2016 [35]
Prognosis/ PFS and OS	CE MRI	15	RNA-sequencing of tumor and adjacent tumor parenchyma/Transcriptomics	Yes/TCGA and GSE 1456	423/1362	8, 12	2017 [36]
Response to Treatment/ pCR	MRI	10	FISH and IHC/Genetics	No	311/311	2	2020 [37]
** *Ovarian Cancer* **							
Prognosis/OS	CT	9	CCNE1 cyclin E1 gene + CLOVAR transcriptomic profiles/Genetics and Transcriptomics	No	38/38	2	2017 [38]
Prognosis/ PFS and OS	CT	5	BRCA mutation/Genetics	No	88/88	2	2018 [39]
Prognosis/ PFS and OS	CT	20	DNA sequencing of tumors/Genetics	Yes/TCGA	364/364	5	2019 [40]
** *Endometrial Cancer* **							
Prognosis/RFS and DSS	MRI	12	RNA sequencing/Transcriptomics	Yes/TCGA	51/487	2, 7, 10, 12	2021 [41]
** *Renal Cell Carcinoma* **							
Prognosis/MFS and OS	CT	12	Gene expression/Genetics and Transcriptomics	Yes/TCGA + TCIA	509/520	10, 12	2020 [42]
Prognosis/OS	CT	11	Hypoxia-related genes/Genetics	Yes/TCGA + TCIA	419/419	3, 5, 17,	2021 [43]
Prognosis/PFI	CT	15	Gene mutations and gene expression/Genetics and Transcriptomics	Yes/TCGA + TCIA	78/214	1, 5, 10, 12	2021 [44]
Prognosis/ PFI and OS	CT	6	Differentially expressed genes and enriched pathways/Transcriptomics	Yes/TCGA + Array-Express database	261/261	5	2021 [45]
Prognosis/ OS	CT	14	mRNA subtype, miRNA subtype, VHL mutation/Genetics and Transcriptomics	Yes/TCIA	267/443	2, 12	2021 [46]
Prognosis/ OS	CT	14	Mutated genes (VHL, BAP1, PBRM1, SETD2), four mRNA patterns/Genetics and Transcriptomics	Yes/TCGA + TCIA	279/382	12,19	2021 [47]
** *Bladder Cancer* **							
Prognosis/ PFI	21	8	RNA sequencing/Transcriptomics	Yes/TCGA + TCIA	62/62	1, 2	2019 [48]
** *Prostate Cancer* **							
Risk Assessment/Risk of metastasis	MRI	10	RNA expression/Transcriptomics	No	64/64	1, 2, 5	2019 [49]
** *Esophagus Cancer* **							
Treatment Response/pCR	CT	17	RNA sequencing/Transcriptomics	No	40/231	1, 12	2020 [50]
Prognosis + Treatment Response/DFS, OS and pCR	CT	17	Gene expression/Transcriptomics	No	28/106	1, 7, 12	2021 [51]
** *Gastric Cancer* **							
Prognosis + Treatment Response/ PFS and OS	CT	11	RNA sequencing/Transcriptomics	Yes/ TCIA	47/475	12	2021 [52]
** *Colorectal Cancer* **							
Prognosis/ OS	CT	8	BRAF mutation/Genetics	No	145/145	10	2020 [53]
** *Hepatocellular Carcinoma* **							
Risk Assessment/Early recurrence	MRI	7	Gene expression/Transcriptomics	No	48/48	1, 2, 10	2020 [54]
** *Melanoma* **							
Prognosis/ OS	FDG PET-CT	7	RNA sequencing. Whole-exon sequencing for common oncogenes. BRAF mutation/Genomics and Transcriptomics	No	33/52	2, 3, 7, 12	2021 [55]
Solid tumors *							
Prognosis + Treatment Response/ PFI and OS	CT	9	Gene expression (CD8 cells)/Transcriptomics	Yes/TCGA	254/491	3, 4, 8, 12	2018 [56]

Abbreviations: CGGA: Chinese Glioma Genome Atlas, CT: computerized tomography, DCB: Durable clinical benefit, DFS: Disease-free survival, DSS: Disease-specific survival, FDG: fluorodeoxyglucose, GDC: Genomic Data Commons, GSE: Gene Expression Omnibus, MFS: Metastasis-free survival, MRI: magnetic resonance imaging, OS: overall survival, pCR: pathological complete response, PET: positron emission tomography, PFI: Progression-free survival, PFS: Progression-free survival, RFS: Recurrence-free survival, TCGA: The Cancer Genome Atlas, TCIA: The Cancer Imaging Atlas, TTF: time-to-treatment failure. * Head and neck-squamous-cell carcinoma (HNSC), lung squamous cell carcinoma (LUSC), lung adenocarcinoma (LUAD), liver hepatocellular carcinoma (LIHC), bladder endothelial carcinoma (BLCA), sarcomas, etc. ** Limitations: 1: High Dimensionality/Small n-to-p Data bias, 2: Lack of external validation, 3: Population bias, 4: Spectrum bias, 5: Disease severity bias, 6: Disease prevalence bias, 7: Selection bias, 8: Detection bias, 9: Test execution variation, 10: Test technology bias, 11: Treatment paradox, 12: Verification bias, 13: Inappropriate reference standard bias, 14: Review bias, 15: Clinical review bias, 16: Incorporation bias, 17: Observer variability, 18: Handling of indeterminate results, 19: Arbitrary choice of a threshold value, and 20: Hot stuff bias. The full list and explanation of limitations and biases evaluated for each study can be found in Appendix B.

**Table 2 ijms-23-06504-t002:** Search syntax for electronic databases.

Database	Syntax
PubMed	(“radiomic *” [All Fields] AND (“genetic *” [All Fields] OR “genomic *” [All Fields])) OR “imaging genomic” [All Fields] filters: human and English language ^1^
EMBASE	(radiomics.mp AND exp genetics/) OR imaging genomics.mp filters: human and English language ^2,3^

^1^ word * meaning search for all word endings. ^2^ Exp word/meaning search as entree word. ^3^ word.mp meaning search in title, abstract, heading word, etc.

## Data Availability

Data available upon reasonable request to the corresponding author.

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
