# Peer review of "What Genetics Can Do for Oncological Imaging: A Systematic Review of the Genetic Validation Data Used in Radiomics Studies"

_ijms, 2022, doi:10.3390/ijms23126504_

Round 1
Reviewer 1 Report
Dear authors,
The proposed manuscript deals with emerging fields that promise to bring significant contributions to clinical practice. I appreciated the thorough analysis of a considerable number of papers that deal with radiomics and radiogenomics. The review follows the criteria of Prisma protocol and provides extensive information on the subject. I believe the review provides a clear picture of the level reached at this moment and can be a good reference for those wanting to overview the body of evidence published on the subject.
I recommend the review for publishing with minor reviews to the English prosody.
Author Response
We thank the reviewer for his/her critical review and valuable comments. We have considered all the recommendations and suggestions. Itemized responses are listed below. All the modifications have been tracked with control changes in the word document throughout the manuscript, but there also is a clean version to make its revision easier (See Revised Manuscript).
Dear authors,
The proposed manuscript deals with emerging fields that promise to bring significant contributions to clinical practice. I appreciated the thorough analysis of a considerable number of papers that deal with radiomics and radiogenomics. The review follows the criteria of Prisma protocol and provides extensive information on the subject. I believe the review provides a clear picture of the level reached at this moment and can be a good reference for those wanting to overview the body of evidence published on the subject.
I recommend the review for publishing with minor reviews to the English prosody.
Answer: We thank the reviewer for his/her insightful comments and we are glad to hear that our work was of interest to you and see the potential of this method in future research. As per his/her recommendation, we have reviewed the manuscript to correct all the grammatical or orthographical errors, and in some sections made small changes to sentences to improve the readability of the paper.

Reviewer 2 Report
After reviewing the manuscript titled “What genetics can do for oncological imaging: A systematic review of the genetic validation data used in radiomics studies” several times, I do not have major concerns about the manuscript. The followings are some minor comments have been pointed out that the authors may want to consider making the manuscript more readable.
1. Line 29 Keywords: The keyword “genetical validation” should be “genetic validation”. I don’t think “cancer genetics” is suitable to be a keyword as it only appears two times in the manuscript. Please consider switch to another one.
2. Line 52: Please provide a higher resolution image to make it looks better and more comfortable.
3. Line 105: The Figure 2 is very hard to read. Provide a higher resolution image.
4. Line 117: Provide a better quality of Figure 3. A higher resolution one might be better.
5. Line 128: The context in Figure 4 is unreadable. The pie is too big while the contexts are too small. Provide a high resolution image.
6. Line 154: The Y-axis is missing. Provide a high resolution image.
7. Line 158: Provide higher resolution images for Figure 6.
8. Line 190: Provide a higher resolution image for Figure 7. The pie is too big.
9. Line 193: What’s the meaning of the numbers under the column of “limitations”? The Table A1 should be cited here to make it clearer.
10. Line 193 Table 1 Lung and Head & Neck Cancers: Why do the first two rows have the same citation but different years?
11. Line 297: A space is needed before the word “and”.
12. Line 319: A space is needed before the word “also”.
13. Line 342: Please use italic p as it refers to a p-value. And homogenous the format with or without a space before and after the signs “<”, or “=”, and so on. Check throughout the manuscript. For example, line 446, and so on.
14. Line 354: Please double-check “p=?”.
15. Line 407: Format should be consistent “P < .0001”. It should be “0.001”, lower case p, and italic.
16. Line 407: Please check “28.Li2016”.
17. Line 428: There is an extra space that should delete before the word “differentiated”.
18. Line 469: Please insert citation to “Zeng et al. (2021)”. Check throughout the manuscript.
19. Line 477: What’s the meaning of “#26}”?
20. Lines 485-489: Please insert citations to this paragraph.
21. Line 538: Please check “{Zhou, 2020 #94]”.
22. Line 551: Please cite the literature correctly, “Negreros-Osuna et al. (2020)”.
23. Line 602: What’s the meaning of “1.æ rts2014”?
24. Lines 658-659: Please cite the literature correctly.
25. Lines 704-707: Please cite the literature correctly.
26. Line 710: Please cite the literature correctly.
Author Response
We thank the reviewer for his/her critical review and valuable comments. We have considered all the recommendations and suggestions. Itemized responses are listed below. All the modifications have been tracked with control changes in the word document throughout the manuscript, but there also is a clean version to make its revision easier (See Revised Manuscript).
After reviewing the manuscript titled “What genetics can do for oncological imaging: A systematic review of the genetic validation data used in radiomics studies” several times, I do not have major concerns about the manuscript. The followings are some minor comments have been pointed out that the authors may want to consider making the manuscript more readable.
Answer: We thank the reviewer’s comments and agree with the comments. We thank the reviewer for such thoughtful insight into our work and appreciate his/her comments. We have thoroughly improved the manuscript's new version with the suggested changes. We have corrected the acronym for SD in all our tables. We have also further explained the age differences between the groups and referred the reviewer to read question number 4. We have also expanded the limitations as suggested.
- Line 29 Keywords: The keyword “genetical validation” should be “genetic validation”. I don’t think “cancer genetics” is suitable to be a keyword as it only appears two times in the manuscript. Please consider switch to another one.
Answer: We thank the reviewer's suggestions, as keywords are very important. We have changed the keywords as suggested, so now they stand as “genetic validation” and “cancer”, respectively.
- Line 52: Please provide a higher resolution image to make it looks better and more comfortable.
Answer: This is a reprint from another article, but we will ask the journal of Radiology for a higher resolution image.
- Line 105: The Figure 2 is very hard to read. Provide a higher resolution image.
Answer: We have redone the figure in a different program to obtain better quality.
- Line 117: Provide a better quality of Figure 3. A higher resolution one might be better.
Answer: Following the reviewer’s suggestion we have tried to improve the resolution of the figures in the article, but this is a little tricky. In our original version the images can be read quite well, but in the PDF version that the reviewers got almost all of the figures are blurry. We can suggest to the editor that at the editing stage we send in TIFF or PNG files of all the original figures to avoid such problems in the quality of the figures.
- Line 128: The context in Figure 4 is unreadable. The pie is too big while the contexts are too small. Provide a high-resolution image.
Answer: The same as number 4.
- Line 154: The Y-axis is missing. High-resolution image.
Answer: We have added the Y-axis to the image and improved the resolution. See answer number 4.
- Line 158: Provide higher resolution images for Figure 6.
Answer: See answer number 4.
- Line 190: Provide a higher resolution image for Figure 7. The pie is too big.
Answer: See answer number 4.
- Line 193: What’s the meaning of the numbers under the column of “limitations”? The Table A1 should be cited here to make it clearer.
Answer: As per the reviewer's suggestion we have cited the limitations table here, but also added the numbers and their corresponding limitations to the legend.
- Line 193 Table 1 Lung and Head & Neck Cancers: Why do the first two rows have the same citation but different years?
Answer: It is because the same first author has published two articles on lung cancer and head and neck cancers in two different years. The first one is one of the first studies to be published in radiomics with the use of genetic data, and the second one is about the response to treatment in cancer patients applying radiogenomics methodology. The reason they had the same reference number was due to an error, it has been corrected.
- Line 297: A space is needed before the word “and”.
Answer: It has been corrected.
- Line 319: A space is needed before the word “also”.
Answer: It has been corrected.
- Line 342: Please use italic p as it refers to a p-value. And homogenous the format with or without a space before and after the signs “<”, or “=”, and so on. Check throughout the manuscript. For example, line 446, and so on.
Answer: Dear reviewer, it has been checked throughout the manuscript and changed accordingly.
- Line 354: Please double-check “p=?”.
Answer: We have double-checked it, and there is no p-value to report with these data, so it has been deleted.
- Line 407: Format should be consistent “P < .0001”. It should be “0.001”, lower case p, and italic.
Answer: Dear reviewer, it has been checked throughout the manuscript and changed accordingly.
- Line 407: Please check “28.Li2016”.
Answer: It was checked and corrected.
- Line 428: There is an extra space that should delete before the word “differentiated”.
Answer: It has been corrected.
- Line 469: Please insert citation to “Zeng et al. (2021)”. Check throughout the manuscript.
Answer: Correct citation was inserted.
- Line 477: What’s the meaning of “#26}”?
Answer: It is an error of the citation program, it has been corrected.
- Lines 485-489: Please insert citations to this paragraph.
Answer: Citations have been introduced as requested.
- Line 538: Please check “{Zhou, 2020 #94]”.
Answer: Correct citation was inserted.
- Line 551: Please cite the literature correctly, “Negreros-Osuna et al. (2020)”.
Answer: Citation has been corrected.
- Line 602: What’s the meaning of “1.æ rts2014”?
Answer: It is an error of the citation program, it has been corrected.
- Lines 658-659: Please cite the literature correctly.
Answer: Citation has been corrected.
- Lines 704-707: Please cite the literature correctly.
Answer: Citation has been corrected.

Reviewer 3 Report
In this systematic review, Mombiela and colleagues provided a comprehensive overview of the progress in the field of Radiogenomics. This is a well-written manuscript with thoroughly researched studies and their critical analysis. The authors also discussed the limitations and advantages in good detail. Overall, it needs just a minor revision which is as below.
- Figure 5: Although already mentioned in the text, it will be more helpful to provide abbreviations for the short forms used in the figure. These can be added either to the figure itself or as a part of the figure legend.
- Figures are a little blur and should be of better resolution.
- For the better flow – I recommend the conclusion to be just after the discussion and before the materials and method section.
- Line 247 – Reference (Kong(2019)) needs to be cited in its correct way
- Line 602 – Reference (aerts(2014)) needs to be cited in its correct way
Author Response
We thank the reviewer for his/her critical review and valuable comments. We have considered all the recommendations and suggestions. Itemized responses are listed below. All the modifications have been tracked with control changes in the word document throughout the manuscript, but there also is a clean version to make its revision easier (See Revised Manuscript).
In this systematic review, Mombiela and colleagues provided a comprehensive overview of the progress in the field of Radiogenomics. This is a well-written manuscript with thoroughly researched studies and their critical analysis. The authors also discussed the limitations and advantages in good detail. Overall, it needs just a minor revision which is as below.
Answer: We thank the reviewer’s comments and fill with joy that our hard work has been appreciated. We agree with the comments and have made the appropriate changes as suggested.
Figure 5: Although already mentioned in the text, it will be more helpful to provide abbreviations for the short forms used in the figure. These can be added either to the figure itself or as a part of the figure legend.
Answer: Following the reviewer’s suggestion we have added the abbreviations to the legend of figure 5.
Figures are a little blur and should be of better resolution.
Answer: Following the reviewer’s suggestion we have tried to improve the resolution of the figures in the article, but this is a little tricky. In our original version the images can be read quite well, but in the PDF version that the reviewers got almost all of the figures are blurry. We can suggest to the editor that at the editing stage we send in TIFF or PNG all the original figures to avoid such problems in the quality of the figures.
For the better flow – I recommend the conclusion to be just after the discussion and before the materials and method section.
Answer: We agree with the reviewer´s suggestion, it does have a better flow if the conclusion goes right after the discussion. We were just following the journal authors’ guidelines, but if the editor agrees with this suggestion we do prefer it, as the reviewers suggested.
Line 247 – Reference (Kong(2019)) needs to be cited in correct way
Answer: The reference is now cited correctly.
Line 602 – Reference (aerts(2014)) needs to be cited in its correct way
Answer: The reference is now cited correctly.
